# An interaction graph approach to gain new insights into mechanisms that modulate cerebrovascular tone
Sergio Dempsey [1] ✉, Finbar Argus [1], Gonzalo Daniel Maso Talou [1,2] & Soroush Safaei [1,2]

Mechanisms to modulate cerebrovascular tone are numerous, interconnected, and spatially dependent, increasing the complexity of experimental study design, interpretation of action-effect pathways, and mechanistic modelling. This difficulty is exacerbated when there is an incomplete understanding of these pathways. We propose interaction graphs to break down this complexity, while still maintaining a holistic view of mechanisms to modulate cerebrovascular tone. These graphs highlight the competing processes of neurovascular coupling, cerebral autoregulation, and cerebral reactivity. Subsequent analysis of these interaction graphs provides new insights and suggest potential directions for research on neurovascular coupling, modelling, and dementia.

Regulation of cerebrovascular tone (CVT) is necessary for healthy neuro-vascular coupling (NVC), cerebrovascular reactivity (CVR), and cerebral autoregulation. Naturally, understanding the mechanisms which modulate CVT will help to study these phenomena in health and disease. Several reviews have presented mechanistic descriptions with success (see refs. 1–3 for recent reviews). Comparing the content of these reviews reveals a varied prioritisation of mechanism detail and coverage. For example, Sweeney et al.[1] described in detail the action of neurotransmitters (NTs) on vascular smooth muscle (VSM) but ignored the effects of NTs on astrocytes, a critical component as discussed in Bazargani et al.[4]. Membrane proteins are described in detail in the VSM but not in other components of the neuro-vascular unit (NVU). Intracellular mechanics were not discussed, and the downstream effects of membrane proteins were simplified to cause dilation or contraction. Simplifying or missing mechanisms were also observed in Hosford et al.[2] and Zhu et al.[3], which is a common occurrence in the literature due to the large number of mechanisms and the limited space available for a detailed description.

The reviews cited above present comprehensive reviews of CVT mechanisms for their specific scope, but the lack of agreement regarding what CVT mechanisms to include highlights that there is no comprehensive picture for CVT regulation. This fragmentation of knowledge about the contributing mechanisms inherently leads to overlooking meaningful contributions, impeding the development of a comprehensive knowledge base. It also increases the risk of selection bias when considering what information is important to a particular application or experimental design. To tackle these issues, a comprehensive resource for CVT modulation pathways should be available. Thus, we reviewed, collated, and designed interaction graphs to present a broad collection of mechanisms to modulate CVT throughout the brain vasculature. We introduce concepts related to the NVU before discussing the regulatory mechanisms within the NVU. Reviewed mechanisms are presented first and foremost as aggregated interaction graphs (IGs) since the mechanisms are described in greater detail in the original work. Then, we discuss insights gained from visualisation, where we apply these IGs to highlight underemphasised mechanisms, observations in modelling, and new hypotheses in dementia. Lastly, we show how IGs can be extended to include new studies and fill in the blanks of existing research.

## The neurovascular unit

The NVU is the collection of cells that regulate brain vasculature connected as described (based on Schaeffer et al.[5]): The blood-filled lumen is surrounded by endothelial cells that form the initial blood-brain barrier. Attached to endothelial cells are layers of VSM cells that control CVT through relaxation and contraction. The VSM is surrounded by interstitial fluid, which is encapsulated by the endfeet of astrocytes. The space between VSM and astrocyte endfeet is called the perivascular space (PVS). The body and branching arms of astrocytes in extravascular tissue are surrounded by extracellular fluid. The branching arms of astrocytes can interact by proximity with neuron-neuron synaptic clefts. Axon terminals can also end directly on astrocytes.

The specific cells that compose an NVU depend on the scale of interest. In particular, VSM cells are included in the NVU at the arteriolar level, and VSMs are replaced by pericytes at the capillary level[6]. Given the difficulty in discriminating pericytes from VSM cells and the active debate about

[1]Auckland Bioengineering Institute, University of Auckland, Level 6/70 Symonds Street, Grafton, Auckland 1010, New Zealand. [2]These authors contributed equally: Gonzalo Daniel Maso Talou, Soroush Safaei. ✉e-mail: sdem348@aucklanduni.ac.nz

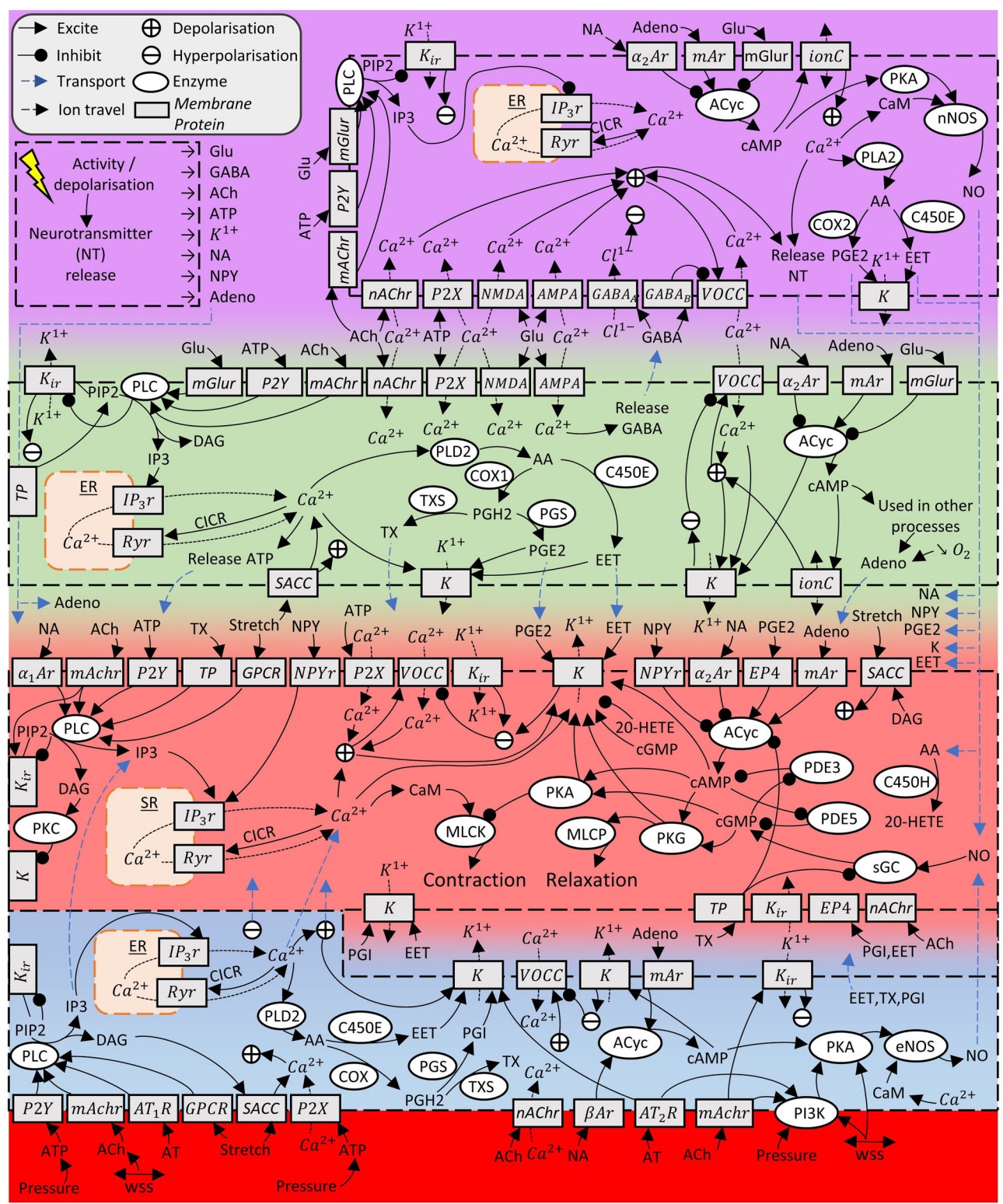

whether pericytes contract at all[1,3,7], there is no basis to propose that pericytes have different mechanisms to modulate CVT if they contract. Therefore, we will consider pericytes mechanisms identical to those of VSM schematically. Future research on the characterisation of pericytes may eventually solidify their identification, but we are still in an early stage to make such a distinction[3]. The type of neuron cells discussed in the NVU also changes depending on its central nervous system location and its expressed neuroreceptors, increasing the NT options for NVC[8]. Lastly, CVT modulation is not limited to one region; by transporting agonists and ions across

endothelial cell membranes, a modulating signal can travel along the vasculature to other territories[9].

## Interaction graphs and analysis of mechanisms modulating CVT

Based on a critical literature analysis we conceptualised IGs for CVT mechanisms. IG1 (Fig. 1.) presents most of the mechanisms collected that have relevance for NVC, CVR, and autoregulation (supporting work in Supplementary Note 1, Table S1 and S2). A separate IG2 (Fig. 2.) highlights

**Fig. 1 | Interaction Graph 1 (IG1): Collected mechanisms of NVC, cerebral autoregulation and CVR.** Solid-coloured dashed boxes represent the neuron (purple), astrocyte (green), VSM (pink), endothelial cell (blue), and vessel lumen (red). Transition colours denote the interface space between cells: purple-green is the synaptic cleft, green-pink is the PVS, and pink-blue is the space between VSM and endothelial cells. Interaction mechanisms are described in Supplementary Note 1. In-figure abbreviations: α-# adrenergic receptor (α#Ar), β-adrenergic receptor (βAr), arachidonic acid (AA), acetylcholine (ACh), adenylyl cyclase (ACyc), adenosine (Adeno), alpha-amino-3-hydroxy-5-methyl-4-isoxazolepropionic acid receptor (AMPAr), angiotensin (AT), angiotensin# receptor (AT#r), adenosine triphosphate (ATP), calcium ($Ca^{2+}$), calmodulin (CaM), cyclic adenosine monophosphate (cAMP), cyclic guanine monophosphate (cGMP), calcium induced calcium release (CICR), cyclooxygenase (1/2) (COX(1/2)), cytochrome P450 (epoxygenase/hydroxylase) (CP450(E/H)), diacylglycerol (DAG), epoxyeicosatrienoic acid (EET), endoplasmic reticulum (ER), E-type prostanoid receptor 4 (EP4), endothelial nitric oxide synthase (eNOS), gamma-aminobutyric acid (GABA), GABA-(A/B) receptor (GABA-(A/B)), glutamate (Glu), G-protein-coupled receptor (GPCR), inositol triphosphate (IP3), IP3 receptor (IP3r), potassium ($K^{1+}$), potassium channels (K), inward rectifying potassium channel ($K_{ir}$), metabotropic adenosine receptor (mAr), metabotropic glutamate receptor (mGlur), muscarinic acetylcholine receptor (mAChr), myosin light chain kinase (MLCK), myosin light chain phosphatase (MLCP), noradrenaline (NA), nicotinic acetylcholine receptor (nAChr), N-methyl-D-aspartate receptor (NMDAr), nitric oxide (NO), neuronal nitric oxide synthase (nNOS), neuropeptide Y (NPY), oxygen gas ($O_2$), phosphodiesterase-3 (PDE3), prostaglandin (E2/I) (PG(E2/I)), prostaglandin synthase (PGS), phosphotidylinositol-3-kinase (PI3K), phosphotidylinositol 4,5-bisphosphate (PIP2), protein kinase A/C/G (PK(A/C/G)), ryanodine receptor (Ryr), soluble guanylyl cyclase (sGC), stretch activated cation channel (SACC), sarcoplasmic reticulum (SR), thromboxane receptor (TP), thromboxane (TX), TX synthase (TXS), voltage operated calcium channel (VOCC), wall shear stress (WSS).

CVT modulating mechanisms of $CO_2$ that have been under-represented in the literature (supporting work in Supplementary Note 2). When any agonist, ion, channel, or enzyme of interest is identified, the cascade of excitation and inhibition effects can be coherently followed to result in contraction or relaxation of the VSM compartment. Each mechanism is based on supporting literature where interactions are described in the Supplementary Material with appropriate citation.

**Interaction graph limitations**

To contextualise interpretation of the IGs, it is important to highlight the limitations of coalescing information from different sources. We include mechanisms discovered in different animal types in our IGs, although they may not all be present in humans. The experiments are from anatomically different locations with varying cell densities and protein expressions, which can alter the magnitude of the mechanism response and change relevant receptor subtypes. Mechanisms are also studied under varying in vivo (e.g., anaesthesia) and in vitro states, which is important as some experimental conditions may not translate to the expected in vivo conditions. The IGs also do not highlight the dependence on age, sex, anatomical location, or the strength of each mechanism. Specified subunits and receptor subtypes are also omitted visually but are included in the references within the Supplementary Material. We placed membrane channels on the side that illustrates a specific mechanistic action, but we do not purport that these membrane proteins are present only in that location. For example, luminally visualised acetylcholine receptors may also be present facing the PVS and interact with perivascular nerves. Lastly, the field is broad, and our collective understanding is far from complete; while we aim to coalesce mechanisms from various fields, mechanisms may still have been missed.

**Analysis**

These IGs allowed us to distil observations that contribute to the state-of-the-art and reaffirm knowledge of the NVU.

**NVC is maintained by several independent pathways that lead to a robust and redundant system (IG1).** The mechanisms presented for NVC form a complicated interconnected web. This redundant network is hypothesised to protect NVC and, by extension, the brain, if a coupling pathway is damaged or inhibited. This supports the review of Hosford et al.[2] in which no studies could completely abolish NVC.

**Adenylyl cyclase and cAMP play an important role beyond the VSM (IG1).** The role of cyclic adenosine monophosphate (cAMP) production within the VSM is recognised[10], somewhat recognised[3] or not recognised at all[1,2,7]. IG1 highlights that several agonists can promote or inhibit production, including purinergic signals, arachidonic acid metabolites, and inhibition of NTs through neuropeptide Y and noradrenaline at the VSM level. This cAMP pathway is also important outside the VSM, where it provides an opportunity for NVC without calcium

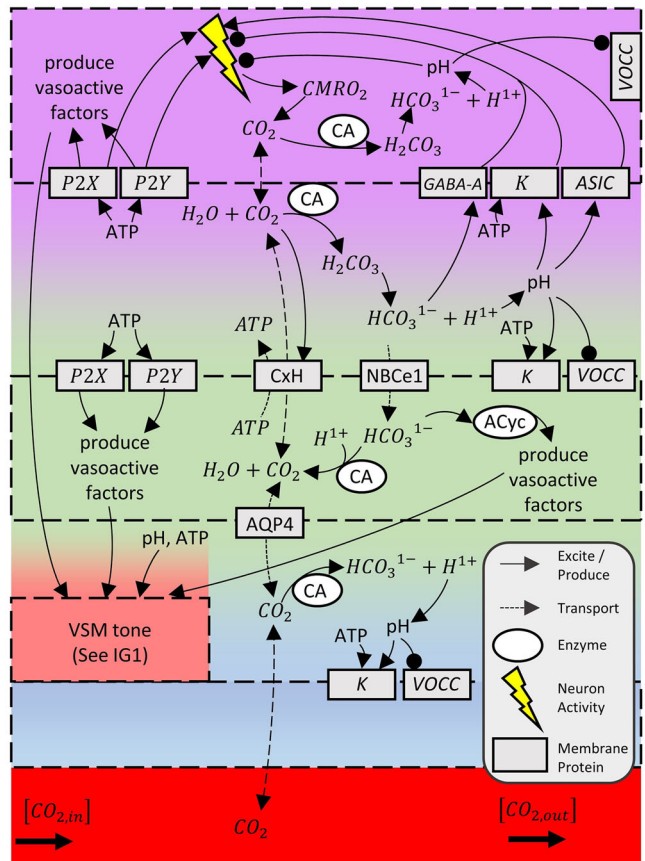

**Fig. 2 | Interaction Graph 2 (IG2): Collected mechanisms of $CO_2$ concerning transport, neural inhibition, and mechanisms of vasomodulation.** Dashed boxes with solid color are: neuron (purple), astrocyte (green), VSM (pink), endothelial cell (blue), and vessel lumen (red). Transition colours denote the interface between cells: purple-green is synaptic cleft, green-pink is PVS, and green-blue is astrocyte to endothelial space. This IG produces vasoactive factors that can be tracked by examining IG1. Interaction mechanisms are described in Supplementary Note 2. In-figure abbreviations: aquaporin 4 (AQP4), acid sensing ion channel (ASIC), carbonic anhydrase (CA), cerebral metabolic rate of oxygen consumption ($CMRO_2$), connexin hemichannel (CxH), electrogenic sodium bicarbonate cotransporter 1 (NBCe1).

transients within astrocytes and neurons by potassium release, which confounds previous conclusions about astrocyte activity in NVC. This was also hypothesised by Bazargani et al.[4].

**Identifying the cAMP role in NVC is needed (IG1).** The dependence on cAMP in NVC is important to know for diseases where NVC is inhibited,

for instance Alzheimer's disease (AD). A recent review (and our updated literature search for years since) shows that no studies have examined inhibiting cAMP/adenylyl cyclase to study NVC directly[2] although studies involving secondary metabolite inhibitors have indirectly involved cAMP reduction (e.g., indomethacin inhibiting prostaglandins which act on adenylyl cyclase[11]. If the effects of cAMP are vital, its importance as a therapeutic target is stressed, as it is already a recognised regulator of blood brain barrier permeability[12].

cAMP reduction is challenging in practice. To this end, we applied our IG1 and identified that activation of $\alpha_2$ adrenergic receptors may have the desired effect on VSM. This would be possible with the administration of dexmedetomidine, which is a selective $\alpha_2$ agonist[13]. Other VSM pathways would remain unchanged for exposure to other stimulating metabolites, allowing the strength of cAMP-independent mechanisms to be measured. This highlights the use of IGs for experimental ideation, though experimental practicalities should also be considered.

**Mechanistic modelling of CVT should follow a reduced systems biology approach (IG1).** There are two popular approaches to modelling the CVT system, experiment-based and following a systems biology modelling philosophy. The former relies on experimentally fitted

coefficients to phenomenological equations to match isolated mechanisms, which can then be combined[14]. Unfortunately, data for all paths in IG1 or IG2 are not available and incur similar limitations needed to interpret our IGs (species, age, location of collected data, etc.), limiting the capability of these models. Systems biology modelling, however, promotes models to match data using reduced modelling that lumps multiple mechanisms together into simple pathways[15]. There is no longer an exact interpretation of the species -- only the overall response to dilate or constrict -- and temporal trends. This approach has been applied for the NVU and three simple vasoactive paths; one contracting and two dilating, have been shown to fit several hemodynamic response functions[16]. This healthy 3-action model agrees with IG1 at the level of contraction and dilation where all sources of CVT modulation are combined into: (1) activation and (2) inhibition of myosin light chain kinase and (3) activation of myosin light chain phosphatase. Thus, the experimentally driven systems biology model was reduced to the same number and type (contracting and relaxing) of final paths in IG1.

**CO₂ provides metabolic feedback mechanisms for NVC (IG2).** Many reviews have disregarded the mechanistic discussion of metabolically produced $CO_2$[1–4,7,10], especially in the context of NVC, because the

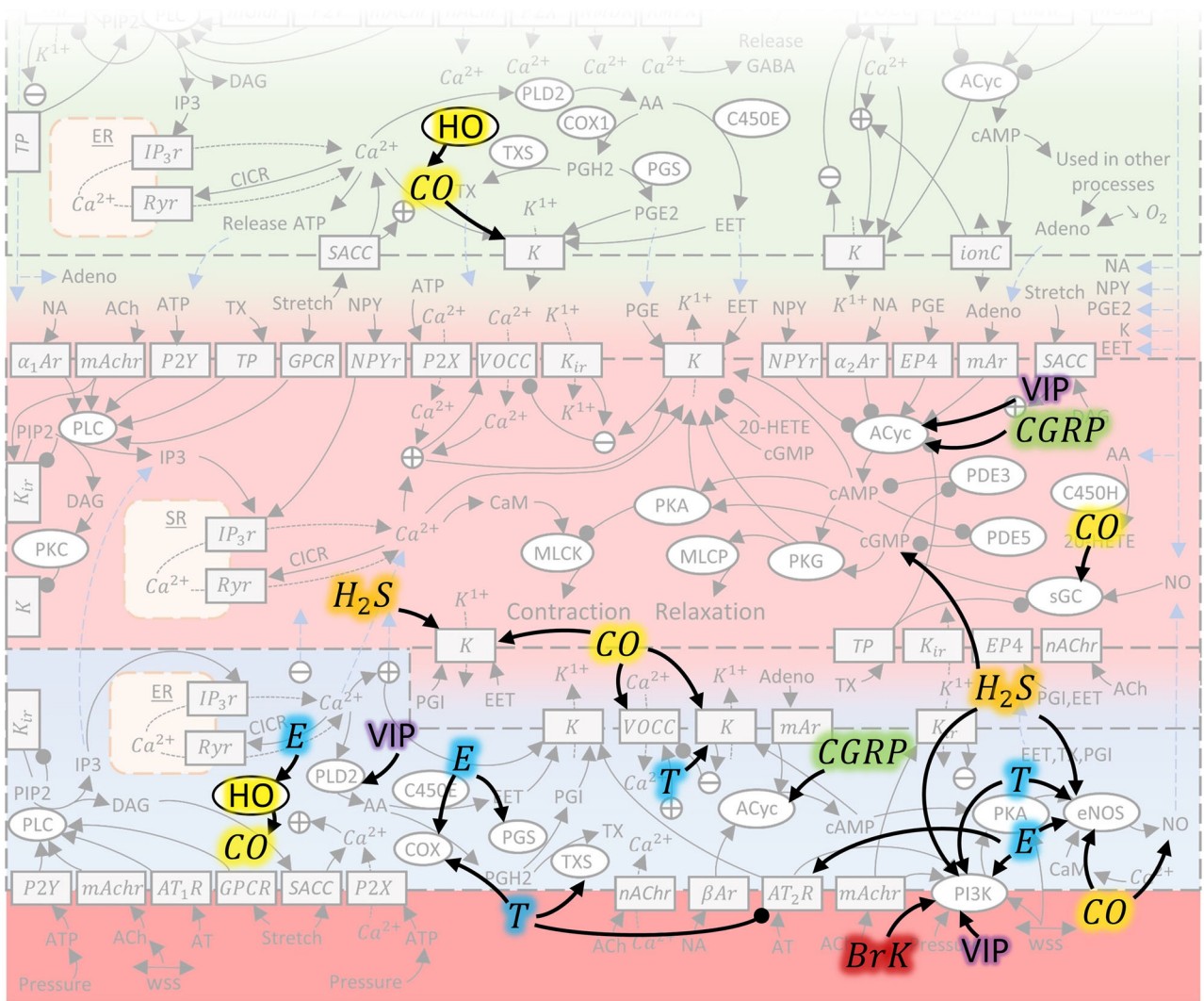

**Fig. 3 | Visualisation of how IGs can be extended and fill in the gaps in pre-existing literature.** Mechanisms included as example are: hydrogen sulfide (orange), carbon monoxide (yellow), perivascular nerve signalling from vasoactive intestinal peptide (purple), and calcitonin gene-related peptide (green), bradykinin (red), and estrogen and testosterone steroids (blue). Interaction mechanisms are described in Supplementary Note 3. In-figure abbreviations: bradykinin (BrK), calcitonin gene related peptide (CGRP), carbon monoxide (CO), estrogen (E), hydrogen sulfide (H₂S), heme-oxygenase (HO), testosterone (T), vasoactive intestinal peptide (VIP).

neuron-astrocyte dynamics are not well understood. However, $CO_2$ is an observed potent neuron inhibitor, vasodilator, and NVC inhibitor[17–19]. We have compiled and emphasised $CO_2$ modulation of CVT from the parenchymal side in IG2 (see Supplementary Note 2 for a complete dynamic description and supporting research). Depending on the strength of inhibition versus vasodilating mechanisms in response to metabolism, IG2 also highlights that $CO_2$ may be a driver of NVC highlighting the importance of continued study.

**$CO_2$ inhibition supports a vascular damage hypothesis for AD (IG2).** In a healthy NVU, inhibition of NVC from $CO_2$ feedback is probably minimal due to efficient $CO_2$ transport to the lumen. However, in diseases such as AD, capillary rarefaction and shuttle impairment are expected[1]. IG2 illustrates that this damage would cause an increase in $CO_2$, which would inhibit neurons, cause vasodilation, and saturate CVT mechanisms, leading to a diminished NVC response. This inhibition may explain cognitive impairment without amyloid and tau positivity, supporting the hypothesis that vascular damage precedes the canonical amyloid cascade hypothesis of AD[20].

**Astrocyte density may prevent cognitive impairment in AD (IG2).** IG2 highlights the important role of astrocytes in $CO_2$ clearance. Here, a higher density of astrocytes means a greater capacity to remove $CO_2$ even with capillary rarefaction or shuttle impairment. This may resist the cognitive impairment symptom of AD, explaining preclinical asymptomatic AD patients following the naming convention of Jack et al.[21]. Interestingly, astrocyte density has also recently been found to help resist amyloid deposition[22]. In recent reviews on astrocytes and dementia, none have mentioned $CO_2$ dynamics or density[23–26], but all discuss astrocytes as an important consideration and target for dementia therapy. The potential role of astrocyte density in the preservation of healthy cognitive function and recent related density research highlight that more studies are needed.

## Learning from, and enhancing IGs

In addition to analysing the IGs themselves, the ability to extend and use this resource is shown by adding several new modulators in IG3 (Fig. 3): perivascular metabolites (calcitonin gene-related peptide, vasoactive intestinal peptide), other gases (carbon monoxide, hydrogen sulfide), steroids (estrogen, testosterone), and bradykinin. References therein are presented in Supplementary Note 3. The emphasis here is that each of these sources only showed a partially complete view of CVT mechanisms. By reading this literature with an IG at hand, once mutual mechanisms were identified, it was clear how and where they fit in the larger picture of CVT, improving interpretation.

## Conclusion

One of the largest issues in the field of mechanistic signalling is proper study design and interpretation when trying to account for hundreds of mechanisms that occur simultaneously at varying strengths. An ideal resource to support this research should highlight mechanisms, their relative strengths, and show experimental conditions (e.g. enzyme knockout) to help understand when (and in what animal/region) a mechanism is important to plan future studies. The static interaction graphs presented here are a first step to drive the generation of such a resource. Mechanisms of NVC, cerebral autoregulation, and CVR that regulate CVT have been collected into holistic IGs. The mechanisms of all three IGs are relevant; separation highlighted the $CO_2$ mechanisms in IG2, and the IG expansion in IG3. We demonstrated the benefit of these IGs in our analysis highlighting new study directions, including the dependence of cAMP on NVC, astrocyte density as a resilience factor to dementia, and highlighting the often-overlooked importance of $CO_2$.

To expand on the functionality mentioned, we are generating an interactive online platform as a resource where researchers can contribute new mechanisms and add supporting research. Any interested reader is encouraged to reach out and join our GitHub community for discussion and contribution (available at https://github.com/ABI-Animus-Laboratory/CVT_IGs). We hope these IGs serve as a jumping-off point for new and established researchers in the field.

### Reporting summary

Further information on research design is available in the Nature Portfolio Reporting Summary linked to this article.

## Data availability

Data sharing not applicable to this article as no datasets were generated or analysed during the current study.

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

## Acknowledgements

All authors gratefully thank the reviewers for fantastic feedback. This work was supported by the Aotearoa Foundation. F.A. is funded by the Andrew Bagnall Fellowship. G.D.M.T. is funded by a Sir Charles Hercus Health Research Fellowship from the Health Research Council of New Zealand. The authors also acknowledge the use of Writefull (https://www.writefull.com/), and Microsoft Word (https://www.microsoft.com/en-nz/microsoft-365/word) for supervised language correction only.

## Author contributions

S.D. Conceptualization, Methodology, Investigation, Writing - Original Draft, Writing - Review & Editing, Visualization F.A. Investigation, Resources, Writing - Review & Editing. G.D.M.T. Resources, Writing - Review & Editing, Supervision, Project administration, Funding acquisition. S.S. Resources, Writing - Review & Editing, Supervision, Project administration, Funding acquisition. G.D.M.T. and S.S. are equally credited co-senior authors in this work.

## Competing interests

The authors declare no competing interests.
