## [Peer review file · Communications Biology]

Reviewers' comments:

Reviewer #1 (Remarks to the Author):

In this perspective the authors aim to summarize from available literature the pathways that regulate vascular tone within the brain and produce a graphical representation of pertinent interactions. The one point of agreement in this field is the complexity of the systems involved and any efforts to summarize them succinctly should be commended. I believe such a resource would be invaluable to the community when interpreting new data and designing experiments. It is especially good to see a detailed inclusion of metabolic considerations in the generation of the NVR that have often been underrepresented in the literature. However, before I recommend publication in *Communications Biology* I have some points of clarification that need to be addressed and some recommendations for the authors.

Structure

I'm unsure that this manuscript, as currently presented, qualifies as a perspective. Usually, a perspective is most effective when they focus narrowly on recent advances in a particular field, highlight areas where knowledge is lacking and suggest potential avenues enquiry. In the present manuscript I see no such conventional outline. While this is more an editorial decision it is my opinion that the authors seem constrained by this format and as a result, clarity is being sacrificed. If their aim is to bring together all known pathways hypothesized to be involved in active control of local brain blood flow then it follows that their discussion of the pathways should be in the main text. Relatedly, their pivot at the end of the manuscript to modelling of the NV response seems a very lengthy way of making the point that future attempts of building a cohesive mathematical model of NVC should consider as many of the pathways identified as possible. Would the authors consider a re-organization of their manuscript to better highlight the strengths of their efforts?

The authors suggest that they will construct an online resource hosted on GitHub that can be added to and edited as the results of new studies emerge. I think this is a major strength of their efforts as the fast-moving field would leave static resources quickly outdated reducing their utility. Could the authors give some more details about this resource in their reply? How will it be implemented? Will there be an active maintainer of the resource or is it to be solely community lead? Some method of searching for and isolating the connections of a particular component of the IG would be most useful.

Content

How did the authors decide on what warrants inclusion in the 'major pathways' depicted in the Figure 1? Was any consideration given to the magnitude of the effect of targeting a specific pathway? Was one study enough? I think this point needs addressing to give the reader some way of evaluating the overall effect on vessel tone of the pathways when activated in concert, a central purpose of collecting this information together.

Figure 1 does not add much for the reader and the anatomical organization in question is mostly indicated in Figures 2 and 3, not to mention its ubiquity in the wider literature. Most interested parties could draw Figure 1 by heart so the description in the text and references given should be sufficient and the space given over to explaining the very complex Figure 2.

The authors discuss the potential importance of cAMP involvement in NVC. Surely, given the multitude of other mechanisms in which cAMP is involved, knock-out studies would not advance the current view of the mechanisms significantly. This complication should be mentioned, at least. Also, indomethacin that has been extensively used to investigate the role of COX in the generation of the NV response is in itself a potent inhibitor of cyclic AMP-dependent protein kinase activity so should be mentioned here.

Regarding the authors interpretation that CO₂ is inhibiting the NVR (Suppl. 2); this overlooks the potential of CO₂ driving (a portion) of the NVR itself. Further, their speculation on the role of Cx26 in this regard is well placed and the mechanism by which they propose (ATP signaling after release

through connexin hemichannels) is supported by fMRI data, see PMID: 25834053. However, Cx26 is not the exclusive connexin that is sensitive to molecular CO₂ so maybe the authors could consider that others could be involved in the mechanism they propose? With respect to the mechanism by which the CO₂ inhibits neuronal activity, they may be interested in this abstract: <https://www.abstractsonline.com/pp8/#!/4649/presentation/34961>

I think that line 163 should read; "...CO₂ increases inhibitory neuron activity, and inhibits the strength..."

Figure 3 retains 'pericyte' while the authors explain earlier their reason to refer to any vessel-associated cells with contractile potential as Vascular Smooth Muscle Cells. Figure 2 does follow this with pink sections indicating VSM. Is there a reason for this?

Patrick Hosford, Tokyo, 15th October 2023

Reviewer #3 (Remarks to the Author):

The stated basic goal of this review was to present a broad collection of mechanisms that modulate cerebral vascular tone (CVT) throughout the brain. The authors state that a comprehensive resource for CVT pathways is needed and that they now present such a broad collection of mechanisms that modulate CVT in this manuscript. The overall goal of the review has potential value and the presentation appears to be new. Having said that, there were significant limitations in what is presented that reduced overall enthusiasm. The stated goal was arguably not achieved even though the authors claim this presentation is state of the art.

More Specific Comments:

1. The authors overstate the 'comprehensive' nature of this review. Some significant details are included. Some of the figures appear to have taken significant effort to prepare, but what is presented is far from comprehensive. There are multiple major mechanisms that regulate vascular tone in brain. These include neurovascular coupling (NVC), autoregulation, chemoregulation, neural and humoral or hormonal regulation, and so forth.
2. The only topic or category that is dealt with in some detail is NVC. Even here, what is presented appears to be based in very large part on NVC mechanisms and cell types within the somatosensory cortex. NVC can vary with brain region and there was no clear integration and similarities or differences in other brain regions such as the frontal (prefrontal) cortex, hippocampus, or cerebellum. Changes in NVC occur with development or maturation, with differences in the relative contribution of prostanoid vs nitric oxide vs CO vs other mechanisms. Some of these are not even considered. Key features of NVC may differ in newborn vs adult vs aged individuals.
3. Autoregulation is mentioned several times in the text, but is not presented in more than a passing manner. The review did not incorporate mechanisms and cell types involved in autoregulation (with either increases or decreases in blood pressure, perfusion pressure) with any real detail. Certainly what is presented for this category of CVT is not comprehensive. Fundamental issues such as are the cellular or molecular sensors of blood pressure changes are not considered. The influence or importance of perivascular nerves (eg, sympathetic nerves) or endothelium on myogenic responses and autoregulation are not included.
4. Other major categories that receive relatively little attention are chemoregulation (effects of CO₂, pH and changes in PO₂ or O₂ delivery), and propagated vasodilation.
5. What is meant by cerebrovascular reactivity in the text, responses to CO₂ or something more broad?
6. Species differences in regulation of vascular tone is a major issue in brain, but is not really considered.
7. What about potential role of other gas transmitters, CO and H₂S?
8. The text states as a disclaimer, these collected mechanisms are based on what has been most important in the literature. How was 'most important' objectively defined?
9. As drawn in Fig 1, the NVU is upstream from capillaries, as no capillaries are shown in this NVU.

There are multiple NVUs, including at the capillary level where other mural cells (pericytes) would be present. These additional mural cells express many receptors and ion channels, and potentially influence capillary diameter and thus blood flow, although this is a controversial area with many positive and negative studies. This area and its controversy is not considered (eg, Trends Neurosci. 2019;42:528-536).

10. While Fig 2 presents a lot of information, specific receptor and ion channel subtypes are not illustrated.

11. The text states that the role of cAMP production within vascular muscle is recognised, somewhat recognized, or not recognised at all. Considerable work on the importance of cAMP and PKA in brain has been done, here are just a few examples: PMID: 35349300, PMID: 11583807, PMID: 8618917.

12. The text states that the role of cAMP in NVC is needed, considerable work related to adenosine has been performed.

13. The text states that endothelial cells regulate tone based on lumen-delivered agonists. If one considered effects of perivascular nerves, interneurons, neurotransmitters and other molecules in the ISF/CSF, etc, there is much more than just agonists in the lumen.

Response to Reviewers

MANUSCRIPT NUMBER: COMMSBIO-23-3277-T

TITLE: RESEARCH DIRECTIONS USING INTERACTION GRAPHS OF CEREBROVASCULAR TONE

First, we thank both reviewers for their insightful comments that show a thorough analysis. The reviews greatly helped us narrow the scope of this manuscript and helped identify a clear goal for continued work where, with community participation, we can create a truly valuable resource.

Both reviewers highlighted that the decision on which “major” mechanisms to include was vague, and we agree. We reworked the manuscript to remove quantitative descriptors. It is difficult to quantify or grade how important a mechanism is, as some mechanisms have been studied far more than others, but that does not mean that the less studied are less important. The magnitude of the mechanism also depends on the experimental conditions and is extremely variable (Hosford et al. 2019).

We have altered the wording to not strongly suggest that we comprehensively cover umbrella topics (autoregulation, general reactivity). In the text, we still refer to the mechanisms that are covered in this umbrella (i.e., CO₂ reactivity). Narrowing the scope of our interaction graphs has not diminished the observations we made, and we are excited that, on continued expansion into new mechanisms of control, more will be learnt by visualising and coalescing all this information.

Both reviewers offered new mechanisms to consider in the interaction graphs. Briefly, we added new mechanisms to the figures (with an updated Supplementary) and we have added an application section including a third IG (IG3) which is an example extension of IG1. We believe that this new figure really highlights how these IGs can be used for new mechanisms. We also included a paragraph on limitations that helped to better interpret the graphs.

This manuscript is intended to highlight that new things can be learnt by collating interaction graphs, reading other work can be enhanced using these graphs, and to stimulate the community to begin a large collaboration where contributors can add new mechanisms (see GitHub discussion) with recognition for their work. We hope that reviewers recognise these goals are achieved with the improved IGs and a more focused discussion based on their feedback.

ITEMIZED RESPONSE TO REVIEWERS

PREAMBLE

In this revised submission, we have included two versions for the main manuscript and supplementary, a clean version with all revisions included but not highlighted and serves as the new manuscript. The Tracked version includes main highlighted text changes based on each reviewer's comments, respectively. The edited text on the comments of reviewer 1 is coloured cyan, and that of reviewer 3 is coloured purple. Each reviewer comment is coded by reviewer number and comment number (e.g., R3C4), which is also tagged in the marked-up text to help explain the change. Line locations are based on the tracked changes pdf where Mline stands for main text line, and Sline stands for supplementary line.

REVIEWER 1 RESPONSE

In this perspective the authors aim to summarize from available literature the pathways that regulate vascular tone within the brain and produce a graphical representation of pertinent interactions. The one point of agreement in this field is the complexity of the systems involved and any efforts to summarize them succinctly should be commended. I believe such a resource would be invaluable to the community when interpreting new data and designing experiments. It is especially good to see a detailed inclusion of metabolic considerations in the generation of the NVR that have often been underrepresented in the literature. However, before I recommend publication in Communications Biology I have some points of clarification that need to be addressed and some recommendations for the authors.

I'm unsure that this manuscript, as currently presented, qualifies as a perspective. Usually, a perspective is most effective when they focus narrowly on recent advances in a particular field, highlight areas where knowledge is lacking and suggest potential avenues enquiry. In the present manuscript I see no such conventional outline. While this is more an editorial decision it is my opinion that the authors seem constrained by this format and as a result, clarity is being sacrificed. If their aim is to bring together all known pathways hypothesized to be involved in active control of local brain blood flow then it follows that their discussion of the pathways should be in the main text...*cont*

- **R1C1** It is difficult to decide how to submit this, and perhaps a perspective is not the right fit. A mini-review/review also suits the work. We believe that it is closer to a perspective for the following reasons: We propose a new method for visualisation and collation that will allow easier development of computational models and future knowledge. We feel that this new method that looks forward to ease future research is less suitable in a typical review paper.

To focus on the visualisation approach, we decided not to include an extensive review of each mechanism within the main text. We feel that this kind of section would not enhance the interpretation of the figure or the proceeding discussion, which is the focus of the work. For the reader who is unfamiliar with a mechanism in the figure, and interested, they can go to the supplementary for details, is the idea.

Relatedly, their pivot at the end of the manuscript to modelling of the NV response seems a very lengthy way of making the point that future attempts of building a cohesive mathematical model of NVC should consider as many of the pathways identified as possible.....*cont*

- **R1C2** We agree the pivot was unexpected and lengthy. We have reformatted this discussion into a single analysis paragraph. It has been significantly shortened.

Would the authors consider a re-organization of their manuscript to better highlight the strengths of their efforts?

- **R1C3** We have reorganised the last section and lightly restructured the document to follow a mini-review format. We feel that this has focused our contribution and, we hope, highlighted the work better. **If the reviewers and editor** feel for the journal that the supplementary should be included in main text (we recognise that most citations would not be formally cited in Supplementary), it will be no trouble to add it as the “mechanism section” without loss of continuity. It would only change the format of a mini review to review to accommodate the manuscript length change. Alternatively, we can also cite all the material from the supplementary in the main document when we refer the reader to the supplementary.

The authors suggest that they will construct an online resource hosted on GitHub that can be added to and edited as the results of new studies emerge. I think this is a major strength of their efforts as the fast-moving field would leave static resources quickly outdated reducing their utility. Could the authors give some more details about this resource in their reply? How will it be implemented? Will there be an active maintainer of the resource or is it to be solely community lead? Some method of searching for and isolating the connections of a particular component of the IG would be most useful.

- **R1C4** Yes. So, the overall plan is 1. Show utility of visualisation, and generate interest (This manuscripts analysis/applications section). 2. Create an interactive resource (isolating, searching) that can store citations and be easily updated (ongoing). 3. With the community that wants to contribute (perhaps you), add and revise mechanistic knowledge as the literature develops. 4. Update periodically in the form of publications when new interpretations become meaningful to the community where contributors are co-authored.

We will develop on SPARC flatmaps. See an example anatomical flatmap here:<https://sparc.science/maps?type=ac>. You can search mechanisms, click and see citation, and using the layers (bottom left toggles) highlight different paths. We are actively working with the developer of these maps to allow named GitHub contributors to annotate paths.

The primary author of this manuscript will be in charge of adding mechanisms to the flatmap until tools for other dedicated editors are developed. The expectation is that the contributor will make a feature request on GitHub with a rough figure of the mechanism + provide citations and other manifest data (type of mechanism, species, location, cell type, etc.). Each feature will be discussed publicly, and on finalisation, the maintainer will add it to the resource.

As a pre-beta, here is the **in-development** CO2 interaction graph: <https://flatmaps.celldl.org/viewer/?id=CO2-demo>. Note: This version is only for driving tool and UI feedback. For the final product, we want community input on ideal type of flatmap (realistically drawn? or just named boxes?) instead of assuming that our format is the best for the community. The anticipated timeline is to have flatmap development completed by September 2024. From whenever this manuscript is published, we will reach out to grow a community for several months, then begin monthly meetings to plan the flatmap shape. Contributor annotation upgrades (Github) should be completed by September 2024.

Sorry if this was lengthy, it is an extremely important question. We are very passionate about creating an impactful resource; more information will be available on GitHub.

How did the authors decide on what warrants inclusion in the ‘major pathways’ depicted in the Figure 1? Was any consideration given to the magnitude of the effect of targeting a specific pathway? Was one study enough? I think this point needs addressing to give the reader some way of evaluating the overall effect on vessel tone of the pathways when activated in concert, a central purpose of collecting this information together.

- **R1C5 (Addressing to both reviewers)** we have removed the wording for quantitative descriptors. Consideration of magnitude of change like Hosford et al. 2019, is difficult in a static figure, but could be emphasised in an interactive figure as a function of number of studies or magnitude of impact. However, with new or single studies, the results may be inaccurate to include. This limitation on displaying the importance of a mechanism, and other limitations regarding interpretation of the figure are addressed in the new limitation discussion (MLines 80-84).

Figure 1 does not add much for the reader and the anatomical organization in question is mostly indicated in Figures 2 and 3, not to mention its ubiquity in the wider literature. Most interested parties could draw

Figure 1 by heart so the description in the text and references given should be sufficient and the space given over to explaining the very complex Figure 2.

- **R1C6** Agreed. We have removed the figure.(MLines 52, 65-66)

The authors discuss the potential importance of cAMP involvement in NVC. Surely, given the multitude of other mechanisms in which cAMP is involved, knock-out studies would not advance the current view of the mechanisms significantly. This complication should be mentioned, at least. Also, indomethacin that has been extensively used to investigate the role of COX in the generation of the NV response is in itself a potent inhibitor of cyclic AMP-dependent protein kinase activity so should be mentioned here.

- **R1C7** Because cAMP has not been directly studied, we think it is interesting to quantify the risk to CVT in pathologies that affect cAMP. The study design would need care, as the reviewer mentioned, given the number of mechanisms that can activate it. We propose a mechanism in the text that could diminish cAMP, while leaving other mechanisms uninhibited for testing. (MLines 114-119)

We were unaware of the indomethacin interaction, and it is surprising that this is not regularly mentioned as a limitation when knocking out COX paths, since more than prostanoid paths would be affected. Another important reason to study cAMP! We have included a relevant discussion in our cAMP highlight point (MLines 110-12).

Regarding the authors interpretation that CO₂ is inhibiting the NVR (Suppl. 2); this overlooks the potential of CO₂ driving (a portion) of the NVR itself...*cont*

- **R1C8** Yes, CO₂ can certainly be interpreted to drive NVC itself, we have raised this point in relevant text.(MLines 143-145)

Further, their speculation on the role of Cx26 in this regard is well placed and the mechanism by which they propose (ATP signaling after release through connexin hemichannels) is supported by fMRI data, see PMID: 25834053...*cont*

- **R1C9** Thank you for sharing this research. We have included that relevant work in the Supplementary Material, where those mechanisms are discussed. (SLines 210-211)

However, Cx26 is not the exclusive connexin that is sensitive to molecular CO₂ so maybe the authors could consider that others could be involved in the mechanism they propose? With respect to the mechanism by which the CO₂ inhibits neuronal activity, they may be interested in this abstract: <https://www.abstractsonline.com/pp8/#!/4649/presentation/34961>

-
- **R1C10** Yes, other connexins are present that are CO sensitive. We originally emphasised Cx26 based on literature suggesting its particular importance. To make this manageable, we have generalised connexins on IG2 and in the Supplementary Material. (SLines 170-220)

Also, we have made it clear that, depending on location, different subtypes of all channels may be available in our limitation section. (MLines 80-81)

I think that line 163 should read; "...CO2 increases inhibitory neuron activity, and inhibits the strength..."

- **R1C11** Yes, thank you for your thorough read, we have amended the text. (SLine 188)

Figure 3 retains 'pericyte' while the authors explain earlier their reason to refer to any vessel-associated cells with contractile potential as Vascular Smooth Muscle Cells. Figure 2 does follow this with pink sections indicating VSM. Is there a reason for this?

- **R1C12** This is a typo from a previous draft when we made that distinction. The figure has now been updated. Main Text Figure 2.

REVIEWER 3 RESPONSE

The stated basic goal of this review was to present a broad collection of mechanisms that modulate cerebral vascular tone (CVT) throughout the brain. The authors state that a comprehensive resource for CVT pathways is needed and that they now present such a broad collection of mechanisms that modulate CVT in this manuscript. The overall goal of the review has potential value and the presentation appears to be new. Having said that, there were significant limitations in what is presented that reduced overall enthusiasm. The stated goal was arguably not achieved even though the authors claim this presentation is state of the art.

- **R3C0** Reviewer 3 raises very important points in detail about the breadth we cover with respect to mechanisms. The mechanisms mentioned by reviewer 3 were fantastic feedback, and this has resulted significant mechanism development, a clear limitations section, and a new interaction graph. The new graph shows how our resource can help explain new mechanisms(or still missing) once the path is found on IG1. Interestingly, in the literature we covered as examples, the discussion was rarely clear on how these factors affected tone (similar to our introduction discussion of unclear simplifications). We felt that using our graphs augmented the understanding of what was presented in the literature of these other mechanisms, which motivated adding that as an application of the graph rather than including them directly on IG1. See Main Text Section 4 (Mlines 164-170), and Supplementary Section 3 (Slines 228-255)

More Specific Comments:

The authors overstate the ‘comprehensive’ nature of this review. Some significant details are included. Some of the figures appear to have taken significant effort to prepare, but what is presented is far from comprehensive. There are multiple major mechanisms that regulate vascular tone in brain. These include neurovascular coupling (NVC), autoregulation, chemoregulation, neural and humoral or hormonal regulation, and so forth.

- **R3C1** Yes, that and more reviewer 3 mentioned impacts CVT (below comments also mentions spatial location, species, age, perivascular nerves, several gasses, and this is not everything). This really highlights how broad the field and mechanisms are.

To improve how comprehensive our figure is, we have added several new mechanisms to IG1, especially in the smooth muscle and endothelium, where, as proceeding comments mention, the mechanisms were limited. (See Main Text yellow highlighted Figure 1 and 2, and Slines 58-63, 107-169)

We also added several new mechanisms on IG3, which is an example of how we used IG1 to learn more about other mechanisms reviewer 3 brought up. We researched CO, H₂S, testosterone, estrogen,

perivascular nerves, O₂ and PO₂, which reviewer 3 mentioned. It turns out that they all fit into the graph quite well, eventually falling into one of the paths on our IG1 which is then mechanistically followed to the VSM. This is shown in the new IG3 showing extensions of IG1. (SLines 228-255, MLines 164-170)

Adding every mechanism to a single figure (IG3 added to IG1) is difficult and likely requires hosting software (see **R1C3**). This has been presented in the future outlooks section which is added in the MiniReview type for Communications Biology.

The only topic or category that is dealt with in some detail is NVC. Even here, what is presented appears to be based in a very large part on NVC mechanisms and cell types within the somatosensory cortex. NVC can vary with brain region and there was no clear integration and similarities or differences in other brain regions such as the frontal (prefrontal) cortex, hippocampus, or cerebellum. Changes in NVC occur with development or maturation, with differences in the relative contribution of prostanoid vs nitric oxide vs CO vs other mechanisms. Some of these are not even considered. Key features of NVC may differ in newborn vs adult vs aged individuals.

- **R3C2** These discussion points are a great example of limitations the community still has today, especially the relative contributions of each which Reviewer 1 also raised which certainly depends on location and age as well. We have included a clear limitations paragraph which discussed these points and others. (MLines 77-91)

Addressing several of these limitations will happen in future work, which is also discussed. (MLines 220-229)

Autoregulation is mentioned several times in the text, but is not presented in more than a passing manner. The review did not incorporate mechanisms and cell types involved in autoregulation (with either increases or decreases in blood pressure, perfusion pressure) with any real detail. Certainly what is presented for this category of CVT is not comprehensive. Fundamental issues such as are the cellular or molecular sensors of blood pressure changes are not considered. The influence or important of perivascular nerves (eg, sympathetic nerves) or endothelium on myogenic responses and autoregulation are not included.

- **R3C3** Thank you for the critique. We read several other mechanisms manuscripts related to autoregulation and myogenic response and added pressure and stretch sensitive mechanisms on the VSM and updated endothelial cell mechanisms in IG1, all mechanisms have been included in the Supplementary Material as well. (SLines 150-166)

Perivascular nerves signalling was addressed in **R3C1** and discussed in IG3, see also the reworked Supplementary Material endothelial cell discussion based on **R3C13** (SLines 123-148) and perivascular nerve discussion for IG3 (SLines 229-237).

Other major categories that receive relatively little attention are chemoregulation (effects of CO₂, pH and changes in PO₂ or O₂ delivery), and propagated vasodilation. Consider highlighting these mechanisms explicitly in the figure/text? Or a section showing what's not been covered in the figure.

- **R3C4** Mechanisms of CO₂, pH are on IG2. In light of this comment and subsequent reading, new mechanisms are now on the endothelial cells in IG2 with an updated Supplementary. (SLines 222-226)

The mechanisms of O₂ are vague in the literature; as a starting point, we have included an increasing amount of adenosine with lower O₂ in IG1. The consensus is that increasing amounts of adenosine with lower O₂ cause a decrease in vascular tone (vasodilates). This has been included through the adenosine - ζ cAMP pathway on IG1.

Mechanisms of propagated vasodilation are beyond the scope of this manuscript, but are briefly discussed in (Mlines 62-64). Ultimately, these are hypothesised to originate as endothelial potassium signaling which is included in our IG1.

What is meant by cerebrovascular reactivity in the text, responses to CO₂ or something more broad?

- **R3C5** Yes the term was vague. We have focused our IG2 specifically on CO₂. CVR is a broad term that likely encapsulates many mechanisms of IG1. We have cleared up our explanation of IG2, to specify that it only relates to CO₂ mechanisms. (MLines 68-70)

Species differences in regulation of vascular tone is a major issue in brain, but is not really considered.

- **R3C6** Yes, great point, this was missed. In line with your previous comment on addressing limitations in the figure, we have included this, along with other assumptions of the mechanisms and presentation. (MLines 78-79)

We have also discussed the options for how the graph can deal with different animal types as future work (see **R3C2**). (MLines 220-229)

What about potential role of other gas transmitters, CO and H₂S?

- **R3C7** We include these mechanisms on IG3 and in the Supplementary Material(see **R3C1**). (SLines 243-248)

The text states as a disclaimer, these collected mechanisms are based on what has been most important in the literature. How was 'most important' objectively defined?

-
- **R3C8** Please see **R1C5**. We have removed any language implying a quantitative methodology for the inclusion of a mechanism.

As drawn in Fig 1, the NVU is upstream from capillaries, as no capillaries are shown in this NVU. There are multiple NVUs, including at the capillary level where other mural cells (pericytes) would be present. These additional mural cells express many receptors and ion channels, and potentially influence capillary diameter and thus blood flow, although this is a controversial area with many positive and negative studies. This area and its controversy is not considered (eg, Trends Neurosci. 2019;42:528-536).

- **R3C9** We address the contention of pericytes in lines (MLines 53-64) and feel that this addresses that controversy. We do mention different types of NVUs, but have enhanced this discussion in the limitation section with added scales, species, and location limitations. (MLines 77-91)

While Fig 2 presents a lot of information, specific receptor and ion channel subtypes are not illustrated.

- **R3C10** Presenting every different subtype of receptors and units is beyond the scope of these figures, as some channels can have more than 30 subtypes. The presence of certain subtypes also has a spatial dependence which we omit in this figure and discuss this in the new limitations section. (MLines - 81,85)

Including spatial information is a great idea to include in the online resource which is now discussed, where different networks can be highlighted for different regions and scales. (MLines 222-224)

The text states that the role of cAMP production within vascular muscle is recognised, somewhat recognised, or not recognised at all. Considerable work on the importance of cAMP and PKA in brain has been done, here are just a few examples: PMID: 35349300, PMID: 11583807, PMID: 8618917.

- **R3C11** We consider cAMP separate to adenosine. We thank for the suggested publications regarding adenosine mechanisms which are now included in our IGs and Supplementary Material. updated discussion is relevant throughout (SLines 140-144) and the endothelial section.

The text states that the role of cAMP in NVC is needed, considerable work related to adenosine has been performed.

- **R3C12** This comment is partially attended in **R3C11**. Adenosine has been studied, but the critical role of cAMP processes that are unaccounted for in other mechanism studies has not been identified.

The text states that endothelial cells regulate tone based on lumen-delivered agonists. If one considered effects of perivascular nerves, interneurons, neurotransmitters and other molecules in the ISF/CSF, etc, there is much more than just agonists in the lumen.

-
- **R3C13** Yes, our endothelium had limited mechanisms. We have significantly updated the endothelium mechanisms and supplementary discussion. (SLines 123-144)

In addition, we have highlighted that receptor placement on IGs is not definite and that these receptors can act with other components of the diagram. (MLines 86-89)

1 REVISED FIGURES

In accordance with revision guidelines we reproduce the revised figures with highlighted changes in yellow.

Reviewers' comments:

Reviewer #1 (Remarks to the Author):

The authors have produced a thorough revision of their original manuscript under a new title 'Mechanisms to Modulate Cerebrovascular Tone: An Interaction Graph Approach'. This revision represents a significant improvement over their initial submission having taken into account all the comments and suggestions from the reviewers. They have also adequately addressed my requests for clarification and concerns in the rebuttal letter, especially regarding the online resource proposed to be attached to this publication. Apart from a few minor points below, I have no further comments or questions regarding this manuscript. I look forward to seeing the interactive IGs when they are available online.

The authors have revised and expanded their IG and explanation section in the supplementary section regarding the mechanisms by which CO₂ interacts with vasculature. The authors highlight pH-sensitive potassium channels as the final target in this IG. It may be worth noting that carbonic anhydrase blockade with acetazolamide does not reduce the magnitude of the neurovascular response shown in PMID: 12843786 and PMID: 35440557 (Suppl. Figure 1 and 2). As they stipulate that carbonic anhydrase hydrating CO₂ as the source of the protons driving the vascular changes, these observations seem to be in contravention to their CO₂ IG. It may well be that sufficient CO₂ hydrations occurs without CA but I think this should be addressed.

There are a few spelling and grammatical errors that need to be corrected before publication. Can I suggest a thorough reading to ensure that these imperfections are caught? The list below may not be exhaustive.

S185: This converted and connexin diffused CO₂ is then diffused through aquaporin-4.... This sentence doesn't make sense and also makes diffusion sound like an active process.

S126: Like all other cells in the NVU, purinergic P2X and P2Y receptors are available to respond to ATP increasing Ca²⁺

S216: VSM also explaining CO₂'s potent vasodilatory effect....

S236: Perivascular nerves may be innervated on....

Reviewer #3 (Remarks to the Author):

In response to the previous review, the authors have addressed most issues raised. In general, the changes have improved the manuscript. One issue was not addressed adequately, plus there are minor issues with spellings etc.

More Specific Comments:

1. In relation to vasomotor responses to CO₂ (hypercapnia) in brain, some new discussion is included. However, the following aspects are not well integrated but relevant considering all the emphasis on cell type etc. Multiple labs have provided evidence that hypercapnia induced vasodilation is not mediated via endothelial cells, rather it is activation of nNOS and acid sensing ion channels (ASIC) in neurons that appears to make the key contribution at lower (more physiological levels of hypercapnia), although with very high levels of CO₂, multiple mechanisms and cell types may be involved (eg, PMID 8512016, 7545691, 9038979, 8967422, 31451088, 7929657).

2. In the abbreviation table, using 'Bre' for bradykinin is confusing. K is not 'potassium'. NO is NOT 'nitrous oxide'. PGS is not 'prostaglandin'.

3. In the text of the supplement (page 3), 'L-arginine' and 'oxygen' are not correct. On page 4, it

is not 'vasodilating'.

Response to Reviewers

MANUSCRIPT NUMBER: COMMSBIO-23-3277A

TITLE: MECHANISMS TO MODULATE CEREBROVASCULAR TONE: AN INTERACTION GRAPH APPROACH

PREAMBLE

We are glad that both reviewers were generally satisfied with our revisions in the previous round. The new comments have been addressed, including a thorough proof reading. In general, the primary text content of the manuscript remained unchanged beyond minor grammar correction. The second interaction graph was updated with some membrane channels and the endothelial cell was largely cleared for hypercapnia-related mechanisms.

- Grammatical errors are corrected in red throughout primary and supplementary manuscript.
- Both reviewers highlighted some issues with IG2 and the supporting supplementary discussion. We greatly revised the structure and wording of this section to significantly improve the clarity and effect of pH. We also amended the interaction graph (IG2).
- Based on Editor suggestions, the manuscript was returned to a perspective format, which resulted in very minor formatting changes in the main manuscript.

Like in the previous round, in this revised submission, we have included two versions for the main manuscript and supplementary work, and a "clean" version with all revisions included but not highlighted serves as the new manuscript. The "Tracked" version includes highlighted text changes in red. Line locations are based on the tracked changes pdf where M#-# stands for main text line range, and S#-# stands for supplementary line range.

ITEMIZED RESPONSE TO REVIEWERS

REVIEWER 1 RESPONSE

The authors have produced a thorough revision of their original manuscript under a new title 'Mechanisms to Modulate Cerebrovascular Tone: An Interaction Graph Approach'. This revision represents a significant improvement over their initial submission having taken into account all the comments and suggestions from the reviewers. They have also adequately addressed my requests for clarification and concerns in the rebuttal letter, especially regarding the online resource proposed to be attached to this publication. Apart

from a few minor points below, I have no further comments or questions regarding this manuscript. I look forward to seeing the interactive IGs when they are available online.

- **R1C0** Thank you again for the thoughtful feedback. We hope you are interested in contributing when the resource is up!

The authors have revised and expanded their IG and explanation section in the supplementary section regarding the mechanisms by which CO₂ interacts with vasculature. The authors highlight pH-sensitive potassium channels as the final target in this IG. It may be worth noting that carbonic anhydrase blockade with acetazolamide does not reduce the magnitude of the neurovascular response shown in PMID: 12843786 and PMID: 35440557 (Suppl. Figure 1 and 2). As they stipulate that carbonic anhydrase hydrating CO₂ as the source of the protons driving the vascular changes, these observations seem to be in contravention to their CO₂ IG. It may well be that sufficient CO₂ hydrations occurs without CA but I think this should be addressed.

- **R1C1** It was misleading to include only 1 membrane channel on VSM. By no means did we mean that pH-dependent K channels are the final action in this IG; all arrows collecting on the VSM are meant to then be considered on IG1. We have removed the potassium channel with pH sensitivity on IG2 in the VSM, and more clearly defined that all sources can contribute. We have also restructured the IG2 section to more clearly define contributions.

There are a few spelling and grammatical errors that need to be corrected before publication. Can I suggest a thorough reading to ensure that these imperfections are caught? The list below may not be exhaustive.

- **R1C2** Absolutely, R3 also found different grammar errors. We thoroughly read the manuscript and made the appropriate changes **visible as red edits**.

S185: This converted and connexin diffused CO₂ is then diffused through aquaporin-4. . . . This sentence doesn't make sense and also makes diffusion sound like an active process.

- **R1C3** Agreed, the phrase has been reworded, "Astrocytic CO₂ then diffuses through aquaporin-4 channels" S197-198

S126: Like all other cells in the NVU, purinergic P2X and P2Y receptors are available to respond to ATP increasing Ca²⁺

- **R1C4** "This ATP opens P2X, and activates P2Y, increasing Ca²⁺..." S132-133

S216: VSM also explaining CO₂s potent vasodillatory effect. . . .

-
- **R1C5** This sentence has been removed and the discussion in Supplementary 2.2 has been reworded.

S236: Perivascular nerves may be innervated on. . .

- **R1C6** "Perivascular nerves release ". S289

REVIEWER 3 RESPONSE

In response to the previous review, the authors have addressed most issues raised. In general, the changes have improved the manuscript. One issue was not addressed adequately, plus there are minor issues with spellings etc.

- **R3C0** We are glad that you are satisfied with the revision. Your comments and references greatly improved the manuscript.

More Specific Comments: In relation to vasomotor responses to CO₂ (hypercapnia) in brain, some new discussion is included. However, the following aspects are not well integrated but relevant considering all the emphasis on cell type etc. Multiple labs have provided evidence that hypercapnia induced vasodilation is not mediated via endothelial cells, rather it is activation of nNOS and acid sensing ion channels (ASIC) in neurons that appears to make the key contribution at lower (more physiological levels of hypercapnia), although with very high levels of CO₂, multiple mechanisms and cell types may be involved (eg, PMID 8512016, 7545691, 9038979, 8967422, 31451088, 7929657).

- **R3C1** Thank you for the supporting literature. Like in the previous round, all have been thoroughly read, and more reading was done. We have revised the IG2 section in the supplementary and the figure accordingly for clarity, including adding ASICs which improved discussion. Thank you again for the resources.

In the abbreviation table, using 'Bre' for bradykinin is confusing. K is not 'potassium'. NO is NOT 'nitrous oxide'. PGS is not 'prostaglandin'.

- **R3C2** We have replaced Bre with BrK, so as not to clash with the Big Potassium (BK) channel acronym. Other grammar issues have been fixed.

In the text of the supplement (page 3), 'L-arginine' and 'oxygen' are not correct. On page 4, it is not 'vasodilating'.

- **R3C3** Thank you for the sharp eye. We thoroughly proofed the manuscript to correct other grammar as well.

1 REVISED FIGURES

In accordance with the revision guidelines, we reproduce the revised figures with highlighted changes in yellow.

Revised Figure 1 (IG1). Note that we had missed adding nicotinic acetylcholine receptors, which have been added. The new discussion on phosphodiesterase justified inclusion in IG1, and the incorrect inhibition (circle) versus activation (arrow) for muscarinic acetylcholine was fixed.

Revised Figure 2 (IG2). We have included pH-dependent K and VOCC inhibition in the supporting cells. We have also included acid-sensing ion channels on the neuron compartment. Lastly, endothelial concepts have been removed based on supplied and other supporting evidence now discussed in Section 2.2 of supplementary.

REVIEWERS' COMMENTS:

Reviewer #3 (Remarks to the Author):

No further comments.